

# Establishment and verification of a nomogram model for predicting the risk of post-stroke depression

Shihang Luo[1], Wenrui Zhang[2], Rui Mao[3], Xia Huang[4], Fan Liu[1], Qiao Liao[1], Dongren Sun[1], Hengshu Chen[1], Jingyuan Zhang[1] and Fafa Tian[1,5]

[1] Department of Neurology, Xiangya Hospital, Central South University, Changsha, China
[2] Department of Neurology, Xuanwu Hospital, Capital Medical University, Beijing, China
[3] Xiangya Hospital, Central South University, Changsha, China
[4] The First People's Hospital of Huaihua, Hunan, Huaihua, China
[5] Department of National Clinical Research Center for Geriatric Disorders, Xiangya Hospital, Central South University, Changsha, China

Corresponding author
Fafa Tian, 2903329471@qq.com

## ABSTRACT

**Objective**. The purpose of this study was to establish a nomogram predictive model of clinical risk factors for post-stroke depression (PSD).

**Patients and Methods**. We used the data of 202 stroke patients collected from Xuanwu Hospital from October 2018 to September 2020 as training data to develop a predictive model. Nineteen clinical factors were selected to evaluate their risk. Minimum absolute contraction and selection operator (LASSO, least absolute shrinkage and selection operator) regression were used to select the best patient attributes, and seven predictive factors with predictive ability were selected, and then multi-factor logistic regression analysis was carried out to determine six predictive factors and establish a nomogram prediction model. The C-index, calibration chart, and decision curve analyses were used to evaluate the predictive ability, accuracy, and clinical practicability of the prediction model. We then used the data of 156 stroke patients collected by Xiangya Hospital from June 2019 to September 2020 for external verification.

**Results**. The selected predictors including work style, number of children, time from onset to hospitalization, history of hyperlipidemia, stroke area, and the National Institutes of Health Stroke Scale (NIHSS) score. The model showed good prediction ability and a C index of 0.773 (95% confidence interval: [0.696–0.850]). It reached a high C-index value of 0.71 in bootstrap verification, and its C index was observed to be as high as 0.702 (95% confidence interval: [0.616–0.788]) in external verification. Decision curve analyses further showed that the nomogram of post-stroke depression has high clinical usefulness when the threshold probability was 6%.

**Conclusion**. This novel nomogram, which combines patients' work style, number of children, time from onset to hospitalization, history of hyperlipidemia, stroke area, and NIHSS score, can help clinicians to assess the risk of depression in patients with acute stroke much earlier in the timeline of the disease, and to implement early intervention treatment so as to reduce the incidence of PSD.

# INTRODUCTION

Currently, stroke is the second leading cause of death in the world (*Gaete & Bogousslavsky, 2008*), and it is also a significant cause of long-term disability in the middle-aged and elderly. There are more than 2 million new cases of stroke in China every year, and it is the disease with the highest disability rate and life loss among all the diseases in that country (*Yang et al., 2017*). The incidence of stroke is expected to increase further due to the aging population, the continued high incidence of risk factors (such as hypertension, hyperlipidemia, diabetes) and poor management. Although access to overall health services has improved, the availability of stroke specialist care varies across the country, especially in rural areas and remote mountain areas, which is one of the reasons for the poor prognosis of stroke patients (*Wu et al., 2019*). However, in some regions, interventions with low risk of adverse reactions (such as the use of antiplatelet and lipid-lowering drugs), stroke unit care and other effective interventions (for example, inadequate use of intravenous thrombolysis, anticoagulation and decompression in patients with indications) also lead to differences in stroke outcomes between regions.

Post-stroke depression (PSD) is the most common mental disorder after stroke, has a negative impact on the functional recovery, rehabilitation response and quality of life of survivors. In stroke patients, about one-third or more are affected by depression (*Sivolap & Damulin, 2019*), which makes it a serious social and public health problem, so the prevention and treatment of antidepressant is worth studying. However, among the consequences of stroke for survivors, post-stroke depression is the most frequent psychiatric problem. PSD is strongly associated with further worsening of physical and cognitive recovery, functional outcome and quality of life. Moreover, depression negatively affects the patients' ability to engage in rehabilitation therapies, a two-way association between depression and stroke has also been established: stroke increases the risk of depression, but depression is also an independent risk factor for stroke (*Villa, Ferrari & Moretti, 2018*). Approximately one-third of stroke patients may have PSD, but the prevalence of PSD varies from study to study, depending on demographic characteristics, diagnostic criteria, inclusion/exclusion criteria, time after stroke and clinical environment in which patients are examined, while the difference in most studies lies in the lack of a diagnostic standard and a unified diagnostic time for PSD, mentioned in a meta-analysis of 43 studies. About 39–52% of stroke patients were diagnosed with depression during 5-year follow-up, while the prevalence rate at any time in 5–10 years was about 29%. Interestingly, in patients with early depression after acute events, a considerable number recovered in the subsequent assessment (*Lenzi, Altieri & Maestrini, 2008*).

Due to the complexity of the pathogenesis of PSD, it is considered to be caused by social psychological factors, pathophysiological factors and other factors, so that there is not a systematic and reliable clinical treatment for PSD, and the therapeutic effect of PSD patients is usually poor (*Starkstein, Mizrahi & Power, 2008*). PSD is an important factor leading to poor recovery of neurological function after stroke, which not only greatly reduces the recovery of cerebral neurological impairment, but also may aggravate the symptoms of patients after stroke, and greatly affects the ability of daily life and work of

patients. At the same time, the mortality rate of PSD patients is also significantly higher than that of stroke patients without PSD. According to related studies, the mortality rate of stroke patients with depression is more than 10 times higher than that of ordinary patients within 10 years (*Llorca et al., 2015*). In view of the fact that the clinical diagnosis of PSD is often insufficient, and the onset of PSD is often one month or more after stroke, most medical environments also lack enthusiasm for PSD. Compared with stroke without depression, PSD often shows higher mortality, worse neurological recovery, more obvious cognitive impairment and lower quality of life (*Starkstein, Mizrahi & Power, 2008*). Therefore, clinicians are familiar with and master the risk factors of PSD, so it is extremely important to carry out early prevention and treatment for such patients (*Arseniou, Arvaniti & Samakouri, 2011*). However, PSD is affected by a variety of risk factors, such as age, sex, education, occupation, income, smoking, drinking, *etc.*); factors related to social support (marital status, length of stay from onset to hospitalization, lifestyle, number of children, *etc.*); and disease-related factors (stroke site, NIHSS score, hypertension, diabetes, history of hyperlipidemia, *etc.*) (*Rabi-Žikić et al., 2020*; *Schöttke et al., 2020*). Considering so many related risk factors, accurate prediction of PSD tools and early intervention by clinicians is an effective means to improve the prognosis of PSD patients (*Vogel, 1995*). Although many previous literatures have studied the relationship between some risk factors and the occurrence of PSD (*De Man-van Ginkel et al., 2013*), no research has been done to predict the risk of PSD by combining these factors with the nomogram of PSD.

The purpose of our study is to establish a nomogram for risk prediction of post-stroke depression, allowing clinicians to conduct an early PSD risk assessment through clinical factors that are easily available in the early stage of the disease, and then carry out early clinical preventive treatment to effectively reduce the incidence of depression in stroke patients.

## PATIENTS AND METHODS

### Patients

The study was approved by the Medical Ethics Committee of Xiang ya Hospital of Central South University (approval number: 201910842). A total of 156 patients with acute stroke (including ischemic stroke and hemorrhagic stroke) were collected from Xiangya Hospital of Central South University from June 2019 to September 2020. Data from 202 patients were collected from Xuanwu Hospital of Capital Medical University from October 2018 to September 2020. Each subject was scored with Hamilton Depression scale at 1 month and 3 months after the onset of acute stroke. According to the score of the third month, the stroke patients were divided into PSD group and non-PSD group.

Admission criteria: (1) after admission, imaging examination proved that the patient was the first stroke, which conformed to the diagnosis of stroke (*Tay, Morris & Markus, 2021*); (2) according to the DSM-5, it was consistent with the diagnosis of depression (*Medeiros et al., 2020*); (3) age between 18 and 80; (4) the time from stroke onset to hospitalization is not more than 14 days. Exclusion criteria: (1) greater than 80 years of age; (2) had a history of mental illness; (3) had severe language disorder and disturbance of consciousness; and (4) other major diseases (such as cancer) were diagnosed during follow-up.

Assessment criteria of patient-related risk factors, and clinical characteristics of the study subjects with and without PSD were summarized in Table 1.

## Statistical analysis

### Screen predictors

Through the LASSO regression analysis of R language, we analyzed the screened factors by multifactor logistic regression analysis, and finally screened out six predictive factors with modeling potential. These factors are independent influencing factors of PSD ($P < 0.05$).

### Establishment of nomogram prediction model

According to the final results of multifactor analysis, the ratio of each risk factor to PSD (OR) was calculated and expressed as 95% confidence interval, and the risk prediction nomogram model of PSD was established by R software.

### Validate PSD risk prediction model

We used a bootstrap repeated sampling for internal bootstrap verification and external verification using the data of 156 patients in Xiangya Hospital of Central South University.

### Drawing decision curve

Finally, the decision curve was used to evaluate the clinical predictive value of the prediction model. By quantifying the net income under different threshold probabilities in the queue, the decision curve was analyzed to determine the clinical validity of the nomogram previously established.

## RESULT

We used LASSO regression to select predictive factors to determine the risk factors associated with PSD from the collected patient data. Due to the large number of predictive factors, and this study is carried out in a small sample size, so LASSO regression analysis is selected to screen the most capable predictive factors. LASSO regression analysis was first proposed in 1996, this method is a kind of compressed estimation. By constructing a penalty function, it gets a finer model, compresses some coefficients, and sets some coefficients to zero. Therefore, the advantage of subset contraction is retained, which is a biased estimation for dealing with collinear data. The advantage of LASSO regression is to filter variables and adjust their complexity while fitting the generalized linear model. Therefore, no matter whether the predictive factors are continuous variables or binary or multivariate discrete variables, we can use LASSO regression to model and then predict. The variable filtering here selectively puts the variables into the model in order to obtain better performance parameters, but in this study, we first carried out single factor regression analysis, selected seven predictive factors, and then put these predictive factors into the model.As shown in Fig. 1 is binomial deviance, The lowest point of the curve is the optimal parameter lambda, Fig. 2 shows the LASSO regression coefficient map of all 19 risk factors, each curve corresponds to a risk factor, where the ordinate is the regression coefficient of the predictor, and the abscissa is log (lambda).

We used multivariate logistic regression analyses to calculate the ratio of each risk factor to PSD, (OR), 95% confidence interval (95% CI) and *P* value ($P < 0.05$. As shown in Table 2,

**Table 1   The clinical influencing factors of whether stroke patients have depression after 3 months.**

| Clinical characteristics | XW Hospital | | XY Hospital | |
|---|---|---|---|---|
| | **PSD (%)** | **N-PSD (%)** | **PSD (%)** | **N-PSD (%)** |
| Sex | | | | |
| Female | 34(69.4) | 102(66.7) | 40(72.7) | 66(65.3) |
| Male | 15(31.6) | 51(33.3) | 15(27.3) | 35(34.7) |
| Age | | | | |
| ≥ 60 years old | 29(59.2) | 86(56.2) | 23(41.8) | 42(41.6) |
| <60 years old | 20(40.8) | 67(43.8) | 32(58.2) | 59(58.4) |
| Education level | | | | |
| Primary or below | 15(30.6) | 68(44.4) | 13(23.6) | 25(24.8) |
| Junior high school | 22(44.9) | 68(44.4) | 29(52.8) | 55(54.4) |
| College or above | 12(24.5) | 17(11.2) | 13(23.6) | 21(20.8) |
| Occupation | | | | |
| Farmer | 19(38.8) | 98(64.0) | 18(32.7) | 33(32.8) |
| Civil servant or staff | 16(32.6) | 26(17.0) | 23(41.8) | 32(31.7) |
| Service industry | 9(18.4) | 13(8.5) | 10(18.2) | 21(20.8) |
| Others | 5(10.2) | 16(10.5) | 4(7.3) | 16(15.8) |
| The mode of work | | | | |
| Physical labor | 3(6.1) | 2(1.3) | 35(63.6) | 53(52.5) |
| Mental labor | 22(44.9) | 32(20.9) | 15(27.3) | 29(28.7) |
| Physical labor is equal to mental labor | 24(49.0) | 119(44.8) | 5(9.1) | 19(18.8) |
| Marital status | | | | |
| Married | 43(87.8) | 141(92.2) | 49(89.1) | 92(91.1) |
| Unmarried | 1(2.0) | 0() | 2(3.6) | 2(2.0) |
| Divorced or widowed | 5(10.2) | 12(7.8) | 4(7.3) | 7(6.9) |
| Number of children | | | | |
| 0 | 1(2.0) | 1(0.7) | 2(3.6) | 3(3.0) |
| 1 | 24(49.0) | 45(29.4) | 17(30.9) | 24(23.7) |
| 2 or more | 24(49.0) | 107(69.9) | 36(65.5) | 74(73.3) |
| Smoking history | | | | |
| Quit smoking | 5(10.2) | 17(11.1) | 5(9.1) | 9(8.9) |
| Less than 20 cigarettes per day | 9(18.4) | 39(25.5) | 16(29.1) | 22(21.8) |
| More than 20 cigarettes per day | 1(2.0) | 34(22.2) | 11(20.0) | 24(23.8) |
| No smoking | 34(69.4) | 63(41.2) | 23(41.8) | 46(45.5) |
| Drinking history | | | | |
| Drinking | 4(8.2) | 16(10.4) | 5(9.1) | 14(13.9) |
| A small amount | 4(8.2) | 26(17.0) | 16(29.1) | 26(25.7) |
| Drinking more than 50g per day | 2(4.1) | 9(5.9) | 12(21.8) | 11(10.9) |
| No drinking | 39(79.5) | 102(66.7) | 22(40.0) | 50(49.5) |

**Table 1** (*continued*)

| Clinical characteristics | XW Hospital | | XY Hospital | |
|---|---|---|---|---|
| | **PSD (%)** | **N-PSD (%)** | **PSD (%)** | **N-PSD (%)** |
| Diabetes history | | | | |
| None | 39(79.5) | 130(85.0) | 40(72.8) | 67(66.3) |
| With poor or untreated treatment | 3(6.2) | 4(2.5) | 11(20.0) | 20(19.8) |
| With good treatment | 6(12.3) | 18(11.8) | 2(3.6) | 10(10.0) |
| Do not know there is a history of diabetes | 1(2.0) | 1(0.7) | 2(3.6) | 4(3.9) |
| History of hypertension | | | | |
| None | 15(30.6) | 47(30.7) | 19(34.5) | 28(27.2) |
| With poor or untreated treatment | 3(6.2) | 33(21.6) | 14(25.5) | 41(40.1) |
| With good treatment | 31(63.2) | 73(47.7) | 19(34.5) | 23(22.8) |
| Do not know there is a history of diabetes | 0(0) | 0(0) | 3(5.5) | 9(8.9) |
| History of hyperlipidemia | | | | |
| None | 42(85.7) | 142(92.8) | 25(45.5) | 37(36.6) |
| Yes | 7(14.3) | 11(7.2) | 30(54.5) | 64(63.4) |
| TIA history | | | | |
| None | 48(98.0) | 152(99.3) | 48(87.3) | 92(91.1) |
| Yes | 1(2.0) | 1(0.7) | 7(12.7) | 9(8.9) |
| The history of operation | | | | |
| None | 39(79.5) | 130(85.0) | 41(74.5) | 68(62.4) |
| Yes | 10(20.5) | 23(15.0) | 14(25.5) | 33(37.6) |
| History of interest | | | | |
| None | 44(89.8) | 144(94.1) | 26(47.3) | 56(55.4) |
| Yes | 5(10.2) | 9(5.9) | 29(52.7) | 45(44.6) |
| The time from onset to hospitalization | | | | |
| Less than or equal to 3 days | 36(73.5) | 128(83.6) | 26(47.3) | 42(41.6) |
| >3 days less than or equal to 7 days | 7(14.3) | 18(11.8) | 21(38.2) | 42(41.6) |
| >7 days less than or equal to 11 days | 6(12.2) | 7(4.6) | 5(9.1) | 6(5.9) |
| >11ays | 0(0) | 0(0) | 3(5.4) | 11(10.9) |
| Stroke area | | | | |
| Anterior circulation | 39(79.6) | 131(85.6) | 43(78.2) | 74(73.3) |
| Posterior circulation | 5(10.2) | 20(13.1) | 11(20.0) | 26(25.7) |
| Anterior and posterior circulation are involved | 5(10.2) | 2(1.3) | 1(1.8) | 1(1.0) |
| NIHSS score | | | | |
| 0–1 point | 17(34.6) | 82(53.6) | 22(40.0) | 33(32.7) |
| 2–4 point | 16(32.7) | 48(31.4) | 20(36.4) | 31(30.7) |
| 5–15 point | 16(32.7) | 23(15.0) | 13(23.6) | 37(36.6) |
| Total | 49 | 153 | 55 | 101 |
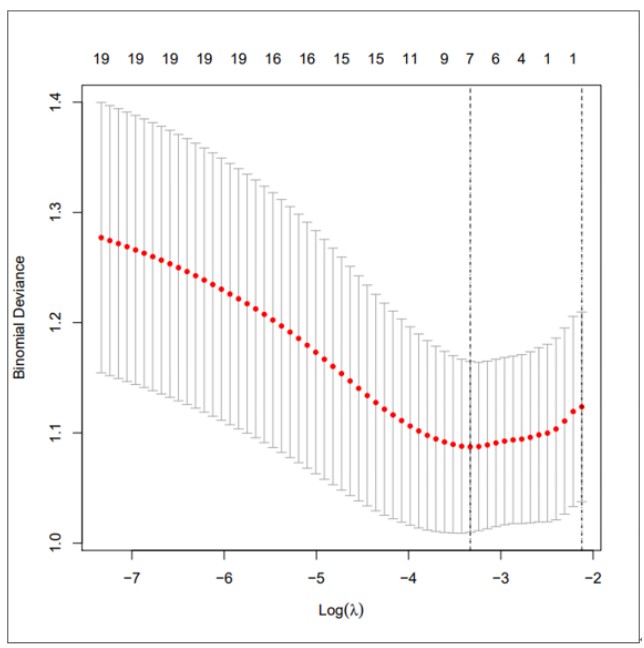

**Figure 1** Preliminary screening of clinical risk factors using LASSO regression.

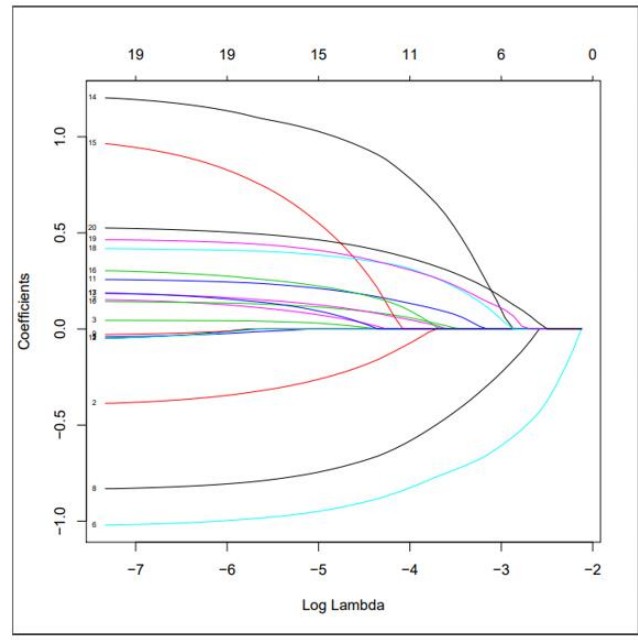

**Figure 2** Use LASSO regression to make a map of risk factor coefficients.

**Table 2  Selection of clinical factors by logistic regression.**

| Influencing factors | Odds ratio | 95% CI | P-value |
|---|---|---|---|
| The mode of work (job-nature) | 0.064 | 0.007–0.470 | 0.007 |
| Number of children (sun) | 0.046 | 0.001–1.415 | 0.048 |
| The time to hospitalization (hospital-stay) | 5.295 | 1.343–11.068 | 0.016 |
| History of hyperlipidemia | 4.803 | 1.331–21.068 | 0.015 |
| Stroke area | 9.592 | 1.339–29.232 | 0.034 |
| NIHSS | 2.747 | 1.020–7.461 | 0.045 |

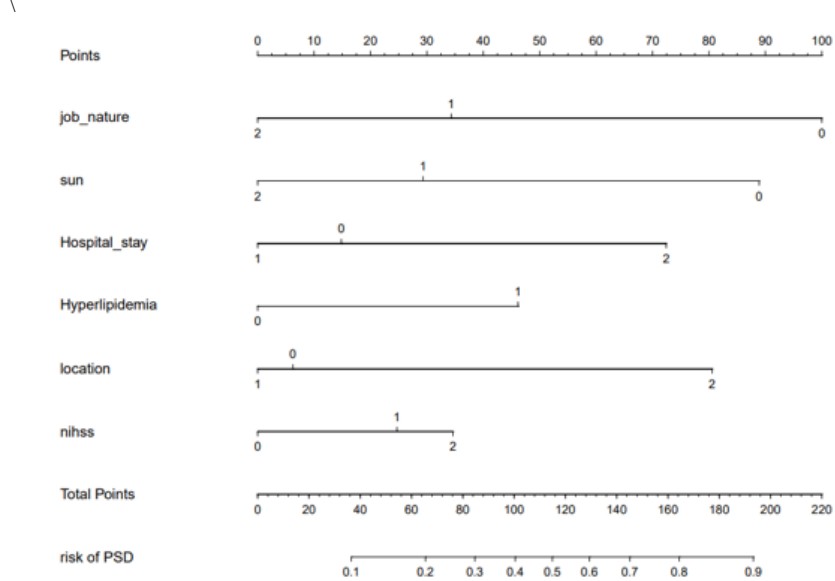

**Figure 3  The nomogram of PSD.**

we found that the P value of drinking history did not reach the standard ($P > 0.05$), so we excluded it.

Using the above prediction factors, we were able to establish the nomogram of PSD risk prediction as shown in Fig. 3. The internal bootstrap verification was carried out by repeated sampling with Bootstrap method, and its C-value was 0.71.

The calibration curve consistency test results of the internal and external verification of the predicted and actual values of our prediction model showed that the PSD risk probability predicted by the nomogram model had a good correlation with the actual PSD risk probability. The results are shown in Figs. 4 and 5.

We further observed that the C-index of internal verification and external verification were 0.773 (95% CI [0.696–0.850]) and 0.702 (95% CI [0.616–0.788]), respectively, indicating that the model had a good ability to predict risk of PSD.

The analysis of the decision curve determines the clinical practicability of the nomogram by quantifying the net income under different threshold probabilities in the cohort. In DCA analysis, the Abscissa represents the threshold probability, that is, the probability
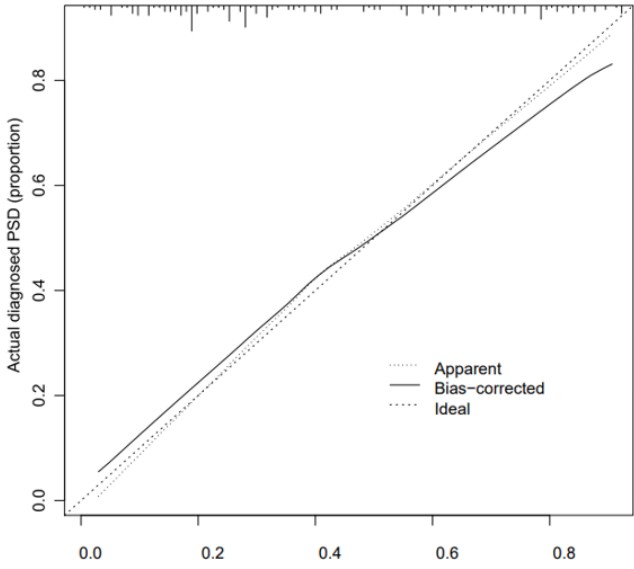

**Figure 4   Correction curve of raining set.**

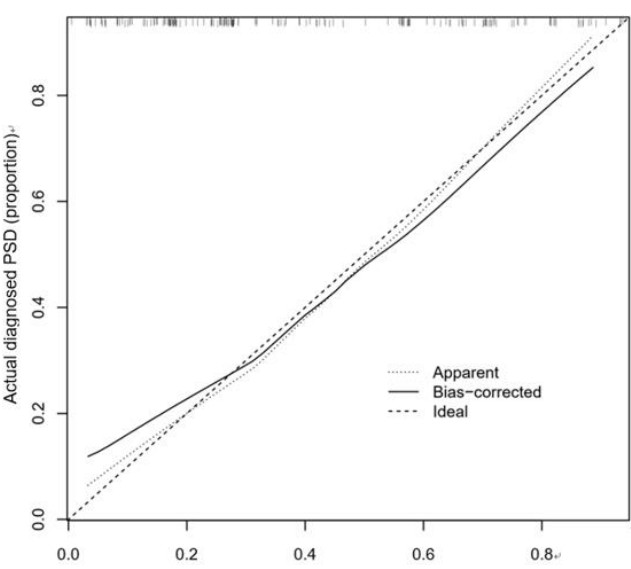

**Figure 5   Correction curve of validation set.**

that the patient will have an outcome event is predicted by the line chart model. When this probability reaches a specified threshold, clinical intervention measures will be taken for stroke patients. At this time, some patients can benefit from the clinical intervention, but there will also be patients who should be treated without intervention or excessive treatment. The ordinate represents the net benefit of the patient after the treatment benefit is subtracted from the treatment loss. Assuming that all patients are negative and all patients

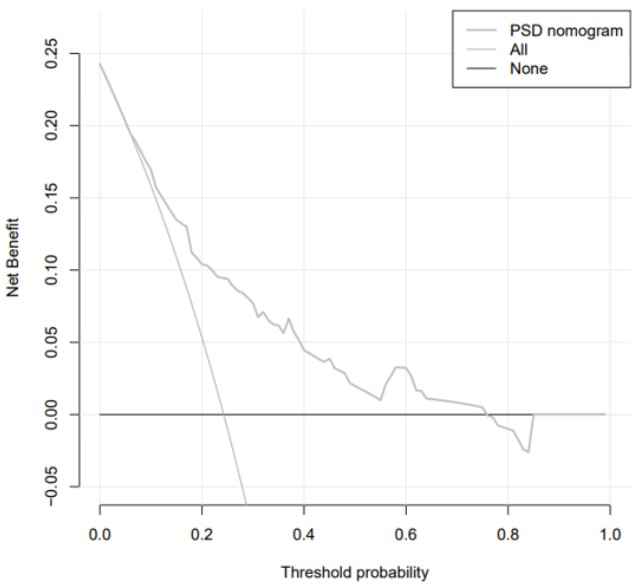

**Figure 6** Decision curve.

do not receive treatment, the net benefit of the patients is 0, showing a line parallel to the X axis on the chart; assuming that all patients are positive and receiving treatment, it is shown as a backslash that intersects the X axis. The curve of the prediction model is distributed between the above two curves, and the farther away from the above two curves shows that the model can get more benefits in a larger threshold range. We observed that when the threshold probability was greater than 6% (Fig. 6), the nomogram was highly practical in clinical practice.

## DISCUSSION

Nowadays, nomograms have been widely used in many medical prognoses such as oncology, and its main advantage lies in its high accuracy and easily comprehensible results, so as to help clinicians to make better clinical decisions (*Iasonos et al., 2008*; *Huang et al., 2016*). In recent years, the incidence of stroke has increased precipitously, seriously endangering the lives of the middle-aged and elderly (*Kao, Chen & Manjunath, 2020*), and the occurrence of post-stroke depression can have a strong negative effect on the prognosis of stroke patients (*Cai et al., 2019*). However, there is almost no direct and effective treatment for post-stroke depression (*Li & Zhang, 2020*). This is also because post-stroke depression is the result of multiple factors and requires a comprehensive, long-term treatment, while early clinical interventions (*Wang et al., 2019*), such as placebo or antidepressant treatment, can greatly reduce the incidence of PSD and improve the prognosis of patients (*Huff, Ruhrmann & Sitzer, 2001*).

Post-stroke depression was recognized by psychiatrists as early as 100 years ago, but systematic case-control studies did not begin until the 1970s (*Robinson & Jorge, 2016*). This is also due to the difficulty of collecting clinical cases of post-stroke depression and the

limitations of late follow-up. Our study is the first to combine the clinical symptoms of patients with living environmental factors, using nomogram to assess the risk of depression in patients with acute stroke. We selected six predictive clinical factors (work style, number of children, time from onset to hospitalization, history of hyperlipidemia, stroke area, and NIHSS score) to develop an easy-to-use nomogram as a new predictive tool for evaluating and predicting the risk of depression after stroke.

These results indicate that the degree of work fatigue not only affects the severity of stroke, increases the physical burden of patients, but also has a great psychological impact on patients (*Volz et al., 2016*); the more tired the work of patients, the greater the psychological burden, leading to a higher incidence of PSD. Similarly, PSD is related to the number of children and the time from onset to hospitalization. We find that type of work, the number of children, and the length of time from onset to hospitalization can be encapsulated as social support factors. Specifically, when the type of work is mainly manual work such as farmers or workers combined with a smaller number of children, and less family support, the easier it is to induce psychological changes. The longer the time from onset to hospitalization means that patients receive less social support, making patients more prone to emotional distress and depression (*Yusrini Susanti, Wardani & Fitriani, 2019*).

PSD is associated with neural regions associated with stroke. We found that patients with indices of stroke in the anterior circulation have a higher risk of depression. This may be because the frontal lobe and temporal lobe of the anterior circulation are significantly related to the occurrence of post-stroke depression (*Price & Duman, 2020*). The frontal lobe is generally considered to play an important role in cognitive and emotional functioning, while the frontal lobe through the frontal-occipital tract pathway may be involved in the occurrence and development of depression (*Howard et al., 2019*; *Nelson et al., 2018*). It mainly depends on the ventromedial prefrontal cortex to process the relevant emotional information. From the point of view of molecular neuropathology, the expression level of mRNA in prefrontal lobe SULT2A, 11 $\beta$-hydrosteroid dehydrogenase and other factors closely related to emotion regulation is significantly increased in patients with severe depression (*Yan et al., 2020*), while the temporal lobe also plays an important role in the regulation of negative emotion. Many clinical imaging studies show that temporal activation in patients with depression is significantly higher compared to healthy subjects during tasks of negative emotion self-regulation (*Maggioni et al., 2019*).

PSD was found to be correlated with the history of hyperlipidemia and NIHSS score. These two risk factors can be summarized as the clinical symptoms of the disease. Usually, the history of hyperlipidemia and the higher NIHSS score often mean that the symptoms of the disease are more serious and the prognosis poor (*Ilut et al., 2017*), leading to greater psychological burden and increased negative emotions, thereby increasing the risk of PSD.

This study combines the clinical data of the two centers, due to the large number of influencing factors, and this study is carried out in a small sample size, so LASSO regression analysis is selected to screen the most predictive factors. The advantage of LASSO regression is to filter variables and adjust their complexity while fitting the generalized linear model. Therefore, no matter whether the predictive factors are continuous variables or binary

or multivariate discrete variables, we can use LASSO regression to model, and then use nomogram to establish a good model to predict the risk of post-stroke depression. At the same time, calibration curve isused to evaluate the model, and it is found that the model has excellent accuracy and differentiation. However, because the quantitative criteria of sample size and influencing factors are different from those in previous studies, some of the results may not be supported. So increasing the sample size of the experiment, as well as more central samples, will increase the persuasiveness of the study, which is what we are doing.

In sum, an accurate risk assessment can help doctors understand the prognosis of patients early and take timely intervention measures (*Almalki et al., 2018*). Our PSD risk prediction model based on patients' clinical factors has a high clinical predictive value, which is helpful for clinicians to prevent early treatment of PSD while reducing the incidence of PSD, and greatly improve the prognosis of stroke patients (*Gu et al., 2020*; *Ramasubbu, 2011*).

### Funding
This work was financially supported by the National Key Research & Development Program of China (grant number 2017YFC1310000). The funders had no role in study design, data collection and analysis, decision to publish, or preparation of the manuscript.

### Grant Disclosures
The following grant information was disclosed by the authors:
The National Key Research & Development Program of China: 2017YFC1310000.

### Competing Interests
The authors declare there are no competing interests.

### Author Contributions

- Shihang Luo conceived and designed the experiments, performed the experiments, analyzed the data, prepared figures and/or tables, authored or reviewed drafts of the article, and approved the final draft.
- Wenrui Zhang performed the experiments, prepared figures and/or tables, authored or reviewed drafts of the article, and approved the final draft.
- Rui Mao performed the experiments, analyzed the data, authored or reviewed drafts of the article, and approved the final draft.
- Xia Huang performed the experiments, authored or reviewed drafts of the article, and approved the final draft.
- Fan Liu performed the experiments, analyzed the data, authored or reviewed drafts of the article, and approved the final draft.
- Qiao Liao performed the experiments, prepared figures and/or tables, and approved the final draft.
- Dongren Sun conceived and designed the experiments, prepared figures and/or tables, and approved the final draft.

- Hengshu Chen performed the experiments, prepared figures and/or tables, and approved the final draft.
- Jingyuan Zhang performed the experiments, prepared figures and/or tables, and approved the final draft.
- Fafa Tian conceived and designed the experiments, performed the experiments, analyzed the data, authored or reviewed drafts of the article, and approved the final draft.

## Human Ethics

The following information was supplied relating to ethical approvals (i.e., approving body and any reference numbers):

The study has been approved by the Medical Ethics Committee of Xiang ya Hospital of Central South University (approval number: 201910842)

## Data Availability

The raw data is available in the Supplemental Files.

## Supplemental Information

Supplemental information for this article can be found online at http://dx.doi.org/10.7717/peerj.14822#supplemental-information.

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
