# Peer review of "Establishment and verification of a nomogram model for predicting the risk of post-stroke depression"

_PeerJ, doi:10.7717/peerj.14822_

## Round 0.1 · original submission · Major Revisions

This manuscript will need major revisions as per suggestion of all 3 peer reviewers.

Reviewer 1 ·

Basic reporting

The manuscript by Luo et al., analyzes the factors forpredicting “PSD”. To build the model, the author first used LASSO to find whichpredictor variables are significant. Then, the selected predictors were furtherevaluated using logistic regression. While the work can be interesting, itsuffers from a significant number of shortcomings that should be addressed.

Majors
1, Line 252-253.“This study combines the clinical data ofthe two centers, due to the large number of influencing factors, and this studyis carried out in a small sample size, so LASSO regression analysis is selectedto screen the most predictive factors.” For model selection, the author used LASSOto find predictor variables with significant contribution to the model. Theselected predictors were further evaluated using logistic regression. >>>>Given that there are 19 predictors (to me,this is not that many compared to >150 observations), could the author run modelselection using “forward selection or stepwise selection” for selecting predictorsfor the model? For example, the author may refer to some similar procedures like thishttp://r-statistics.co/Model-Selection-in-R.html. 

2, The dependent variable “PSD group and non-PSD group” is abinary variable. How accurately can “PSD” be determined? Could the judgement of “PSD”or “not PSD” be different depending on different doctors, given the data werecollected from two separate hospitals? 

3, Since the authors has used “seven predictors” to build theirmodel in the logistic regression, they should show the model in an equation.   

 Minors
1, Abbreviates should be descriptive in full if they appear forthe first time in the text. Such as OR for odds ratio,

Experimental design

The manuscript by Luo et al., analyzes the factors forpredicting “PSD”. To build the model, the author first used LASSO to find whichpredictor variables are significant. Then, the selected predictors were furtherevaluated using logistic regression. While the work can be interesting, itsuffers from a significant number of shortcomings that should be addressed.
Majors1, Line 252-253.“This study combines the clinical data ofthe two centers, due to the large number of influencing factors, and this studyis carried out in a small sample size, so LASSO regression analysis is selectedto screen the most predictive factors.” For model selection, the author used LASSOto find predictor variables with significant contribution to the model. Theselected predictors were further evaluated using logistic regression. >>>>Given that there are 19 predictors (to me,this is not that many compared to >150 observations), could the author run modelselection using “forward selection or stepwise selection” for selecting predictorsfor the model? For example, the author may refer to some similar procedures like thishttp://r-statistics.co/Model-Selection-in-R.html. 2, The dependent variable “PSD group and non-PSD group” is abinary variable. How accurately can “PSD” be determined? Could the judgement of “PSD”or “not PSD” be different depending on different doctors, given the data werecollected from two separate hospitals? 3, Since the authors has used “seven predictors” to build theirmodel in the logistic regression, they should show the model in an equation.    Minors1, Abbreviates should be descriptive in full if they appear forthe first time in the text. Such as OR for odds ratio,

Validity of the findings

2, The dependent variable “PSD group and non-PSD group” is abinary variable. How accurately can “PSD” be determined? Could the judgement of “PSD”or “not PSD” be different depending on different doctors, given the data werecollected from two separate hospitals? 

3, Since the authors has used “seven predictors” to build theirmodel in the logistic regression, they should show the model in an equation.

Reviewer 2 ·

Basic reporting

1. In the introduction section, additional references are needed to support the authors’ argument. For example, in line 51-53, 2 million new cases of stroke; in line 75-76, a meta analysis of 43 studies, etc.

2. In line 68-70, it is hard to follow this sentence to understand the relationship between depression and stroke.

3. In line 191, replace “C value” with “C-index” to make the name of evaluation metric consistent.

Experimental design

1. In line 123-124, the second inclusion criteria is the diagnosis of depression. Since the model is used to predict PSD, why do the authors only include positive samples in the dataset?

2. In line 177, the authors can explain more details on how the net income and clinical predictive value are quantified.

Validity of the findings

1. In the statistical analysis section (line 164-167 and Figure 2), 6 out of 19 predictors are selected. Part of the predictors are categorical variables with multiple levels and it is unclear how the authors perform the selection by LASSO with dummy variables, since the number of coefficients should exceed 19. Can the authors explain?

2. In Table 2, several categorical predictors are finally selected to be used in the logistic regression. But only one point estimate is provided for each categorical variable. For example, the mode of work is estimated with odds ratio 0.064. How to interpret this result given that the mode of work has 3 levels: “physical labor”, “mental labor” and “ physical labor is equal to mental labor”?

3. In discussion section (line 258), the authors mentioned that ROC curve analysis is conducted to evaluate the model, which is not covered in the previous sections.

Reviewer 3 ·

Basic reporting

- The paper is well organized into sections on Data, Statistical Analysis, Results and Discussion, which is very useful for the readers. The quality of English is good throughout. I commend the authors for this manuscript.
- The authors specify the source of data, how long the data was collected for, what kinds of features were present, and how they relate to the classification problem at hand.
- In the section on model building and validation, the authors have listed down their candidate models, and clearly describe how they performed feature selection, tested model performance, and how they arrived at their final ML model.
- Finally, they discuss why this present manuscript will help with predictive diagnosis and early treatment of PSD, thereby bridging between statistical techniques and clinical implications.

Experimental design

- The features selected are intuitively well-founded, and I am happy to see them all listed down in the section on “Patients”. I think that section however should be renamed to “Patient Attributes”.
- In line 124, I am only able to see boxes – which I presume are inequalities? Would urge the authors to fix that.
- While LASSO is probably a good feature selection technique for this problem, I would advise the authors to explain why they are using LASSO instead of, say, explained variance, RF feature importance, etc. in the “Results” section. I see that they have explained this in the “Discussion” section; I recommend bringing this into the results section.
- In Fig. 2, the authors use the letter “lambda” with no reference to what this is – I would imagine this is related to the regularization parameter; the authors need to clarify this in the “results” section, as well as in the caption for Fig 2.
- The authors use AUROC as the performance measure for selecting for final model, as it appears in the discussion. It is not immediately clear how the AUROC is related to the decision curve. I advise the authors to include more discussion in the “Results” section on the connection.
- In Fig. 6, the caption needs to give some more insight into what the plot means. What does +/- benefit mean, etc.

Validity of the findings

- It seems to me that the findings are valid. However, there is a lot that needs to be included in the “results” section.
- The model and feature selection techniques and the performance measures used are sensible and the conclusions derived from them is good.
- As the authors point out, there are several ways to improve the conclusiveness of the present analysis, in particular, more data. With that caveat, I believe the conclusions of the paper are justified.

---

## Round 0.2 · Minor Revisions

Please revise as per comments:

> In the statistical analysis section, the authors used the original LASSO methods (Tibshirani, R. (1996). Regression shrinkage and selection via the lasso. Journal of the Royal Statistical Society: Series B (Methodological), 58(1), 267-288.) to implement variable selection. But the method is not applicable when categorical variables are in the model (See Meier, L., Van De Geer, S., & Bühlmann, P. (2008). The group lasso for logistic regression. Journal of the Royal Statistical Society: Series B (Statistical Methodology), 70(1), 53-71.; Yuan, M., & Lin, Y. (2006). Model selection and estimation in regression with grouped variables. Journal of the Royal Statistical Society: Series B (Statistical Methodology), 68(1), 49-67.), which is the case of this manuscript (E.g. variable Education level, Occupation, Working status, The mode of work, etc.). The reason is that the original LASSO treats the variables independently and hence ignores any relationship between them. It disregards the fact that the dummy variables, taken as a whole, represent the same categorical variable. Can authors modify the framework on variable selection so that it can handle models with categorical variables?

Reviewer 1 ·

Basic reporting

The manuscript by Luo et al., analyzes the factors forpredicting “PSD”. To build the model, the author first used LASSO to find whichpredictor variables are significant. Then, the selected predictors were furtherevaluated using logistic regression.

The authors has adequately addressed my questions.

Experimental design

none

Validity of the findings

none

Additional comments

none

Reviewer 2 ·

Basic reporting

In line 68-70, the relationship between depression and stroke needs further explanations to readers. The authors mentioned a modification in the response letter but there is no updates in the revised manuscript.

Other comments in this section have been addressed.

Experimental design

Comments in this section have been addressed.

Validity of the findings

In the statistical analysis section, the authors used the original LASSO methods (Tibshirani, R. (1996). Regression shrinkage and selection via the lasso. Journal of the Royal Statistical Society: Series B (Methodological), 58(1), 267-288.) to implement variable selection. But the method is not applicable when categorical variables are in the model (See Meier, L., Van De Geer, S., & Bühlmann, P. (2008). The group lasso for logistic regression. Journal of the Royal Statistical Society: Series B (Statistical Methodology), 70(1), 53-71.; Yuan, M., & Lin, Y. (2006). Model selection and estimation in regression with grouped variables. Journal of the Royal Statistical Society: Series B (Statistical Methodology), 68(1), 49-67.), which is the case of this manuscript (E.g. variable Education level, Occupation, Working status, The mode of work, etc.). The reason is that the original LASSO treats the variables independently and hence ignores any relationship between them. It disregards the fact that the dummy variables, taken as a whole, represent the same categorical variable. Can authors modify the framework on variable selection so that it can handle models with categorical variables?

Other comments in this section have been addressed.

Reviewer 3 ·

Basic reporting

- The paper is well organized into sections on Data, Statistical Analysis, Results and Discussion, which is very useful for the readers. The quality of English is good throughout. I commend the authors for this manuscript.
- The authors specify the source of data, how long the data was collected for, what kinds of features were present, and how they relate to the classification problem at hand.
- In the section on model building and validation, the authors have listed down their candidate models, and clearly describe how they performed feature selection, tested model performance, and how they arrived at their final ML model.
- Finally, they discuss why this present manuscript will help with predictive diagnosis and early treatment of PSD, thereby bridging between statistical techniques and clinical implications.

Experimental design

- The features selected are intuitively well-founded, and I am happy to see them all listed down in the section on “Patients and Methods”.
- In line 124, the authors have fixed the issue of typesetting inequalities.
- The authors have now added some explanation on their usage of LASSO and the lambda parameter.
- The authors use AUROC as the performance measure for selecting for final model, as it appears in the discussion. I am glad that they have added some more insight into this.

Validity of the findings

- It seems to me that the findings are valid and the results/discussion section is more comprehensive now.
- The model and feature selection techniques and the performance measures used are sensible and the conclusions derived from them is good.
- As the authors point out, there are several ways to improve the conclusiveness of the present analysis, in particular, more data. With that caveat, I believe the conclusions of the paper are justified.

---

## Round 0.3 · accepted · Accept

Congratulations your article is now accepted by PeerJ.

Reviewer 2 ·

Basic reporting

Comments in this section have been addressed.

Experimental design

Comments in this section have been addressed.

Validity of the findings

I think that the authors have adequately addressed the comments in the revised version of the manuscript. Therefore, I have no further comments.